# Next-Generation Whole-Cell Pneumococcal Vaccine

**DOI:** 10.3390/vaccines7040151

**Published:** 2019-10-16

**Authors:** Victor Morais, Esther Texeira, Norma Suarez

**Affiliations:** Department of Biotechnology, Institute of Hygiene, Faculty of Medicine, University of the Republic, Montevideo 11600, Uruguay; vmorais@higiene.edu.uy (V.M.); texeira.esther@gmail.com (E.T.)

**Keywords:** *Streptococcus pneumoniae*, whole-cell vaccine, RM200, virulence factors

## Abstract

*Streptococcus pneumoniae* remains a major public health hazard. Although Pneumococcal Conjugate Vaccines (PCVs) are available and have significantly reduced the rate of invasive pneumococcal diseases, there is still a need for new vaccines with unlimited serotype coverage, long-lasting protection, and lower cost to be developed. One of the most promising candidates is the Whole-Cell Pneumococcal Vaccine (WCV). The new generation of whole-cell vaccines is based on an unencapsulated serotype that allows the expression of many bacterial antigens at a lower cost than a recombinant vaccine. These vaccines have been extensively studied, are currently in human trial phase 1/2, and seem to be the best treatment choice for pneumococcal diseases, especially for developing countries.

## 1. Introduction

*Streptococcus pneumoniae* is a human pathogen that is responsible for causing pneumonia, otitis media, meningitis, and other infectious diseases. It is a major global cause of morbidity and mortality among children, the elderly, and immune-compromised populations [1].

*S. pneumoniae* are encapsulated, nonsporulating, facultative anaerobic, lancet-shaped Gram-positive bacteria that reside in the human nasopharynx. Based on their thick layer of capsular polysaccharide (CPS), *S. pneumoniae* have been classified into about 97 different specific types according to the Danish classification system [2,3].

One of the most important virulence factors of *S. pneumoniae* is the CPS, which protects the pathogen from host defense mechanisms [4,5]. In addition, virulence factors such as pneumolysin, PspA, and PspC enhance inflammation, inhibit complement activation, and help in surface adherence with the host, among other functions [6,7].

Until now, most clinically isolated *S*. *pneumoniae* have been susceptible to penicillin and other antimicrobial agents, but the global emergence of resistant strains and the fact that, in many cases, the disease can progress faster than the introduction of the antimicrobial treatment, have made pneumococcal disease difficult to treat [8,9]. The large number of different serotypes constitutes one of the main difficulties in the development of an effective vaccine.

### 1.1. Vaccine Development

The first vaccines against pneumococcus were not serotype-specific whole-cell heat-treated vaccines. One of the first registered vaccines was the Pneumo-Bacterin, which was recorded in 1909 in the United States [10]. These non-serotype-specific whole-cell vaccines were distributed until the 1930s with some degree of clinical success in trials performed in South Africa in 1911 by British physician Sir Almroth E. Wright with gold and diamond miners [10,11].

In 1914, Spencer Lister developed a distinct typing system for virulent strains and the first serotype-specific whole-cell pneumococcal vaccines containing serotypes A, B, and C. Nowadays, these serotypes are named 5, 2, and 1, respectively, according to the Danish classification system [10].

Later on, isolation of the CPS, achieved by Dochez and Avery in 1917, was used to identify pneumococcal bacteria serotypes, and it was established as a critical virulence factor [10,12,13,14,15]. From that point, the polysaccharide pneumococcal vaccine was used, and it is still used currently [10].

It was not until 1929 that Avery conjugated pneumococcal polysaccharides to proteins to improve immunogenicity, but this development achieved practical clinical value more than 50 years later with the development of conjugate pneumococcal vaccines [5,16]. In the meantime, only unconjugated polysaccharide vaccines were used.

The introduction of conjugate pneumococcal vaccines (PCVs) has been widely adopted and is used in children in both high- and low-income countries. Recently, Chen and collaborators reported a global modelling analysis of the effect and cost-effectiveness of pneumococcal conjugate vaccination [17]. The authors concluded that these PCVs show established cost-effective protection against pneumococcal diseases in various countries in terms of lives saved and disability averted.

Although vaccination with PCVs can prevent infections due to multiple serotypes of pneumococcus, the World Health Organization (WHO) reported that the universal use of PCVs have provided an opportunity for non-vaccine serotype circulation and serotype replacement to cause an increased proportion of pneumococcal disease in both children and adults [18,19]

However, despite the persistent circulation of vaccine serotypes and serotype replacement and the complexity of the manufacturing processes, the cost of PCVs has decreased in low-income countries due to the pneumococcal vaccine advance market commitment [17].

### 1.2. Pneumococcal Antigens

The CPS is the most surface-accessible structure of *Streptococcus pneumoniae* and resembles an accumulation of viscous material. It is a high-molecular-weight polymer composed of repeating units of oligosaccharides. Capsules of most capsular types are covalently bonded to the cell wall [20]. The CPS helps protect pneumococci from several of the host defense mechanisms [4]. Serotypes differ in both chemical composition and CPS amount. These characteristics are related to the virulence degree, invasiveness, and the survival ability of the bacteria in the circulation of different pneumococcal strains [21,22].

Besides the diversity of the CPS among *S. Pneumoniae* strains, other highly conserved cell proteins contribute to the virulence of pneumococcus [23]. The main proteins and their functions are shown in Table 1. Many of these proteins have been extensively examined as vaccine candidate antigens, because CPS-conjugated protein vaccines are of a specific serotype and do not cover the emergence of new virulent serotypes [3].

The first protein antigens were identified through antibody screening, and specific antibodies were found in children exposed to pneumococcus, namely pneumolysin (Ply), pneumococcal surface protein A (PspA), pneumococcal surface protein C (PspC), and pneumococcal surface antigen A (PsaA) [25]. These proteins were proposed for use in a vaccine with the capacity to induce serotype-independent antibodies and broader protective immunity against pneumococcal infections [31,32].

Ply is a cholesterol-binding protein that is liberated by autolysis of the bacteria and forms pores in eukaryotic membranes [25,33]. Another important effect of Ply is activation of the innate immune system through the TLR4 and NLRP3 inflammasome [25,34,35]. This protein is toxic in its native form, so nontoxic variants have been generated to be used in experimental vaccines [25].

PspA is a choline-binding protein that is attached non-covalently to the cell surface through its C-terminal choline-binding repeat region. The protein is present in all pneumococcal strains and has a variable, but cross-protective, N-terminal region exposed on the surface of the bacteria [36]. Furthermore, PspA has a conserved proline-rich region that has been shown to elicit antibody-mediated protection against invasive pneumococcal disease in mice and has been proposed for use as an antigen in a vaccine [23]. PspA inhibits the activation and deposition of the C3 complement protein on the bacterial surface, binds to lactoferrin, helps to adhere to the surface of the host, and protects bacteria from lactoferrin toxic effects [25,37].

PspC is a similar protein to PspA that is capable of interacting with the complement through C3 and Factor H [25,38,39]. PspC also has adhesion properties through interactions with sIgA and the laminin receptor [25].

PsaA is a conserved lipoprotein that transports Mn^2+^ and also functions as adhesin, which plays a major role in pneumococcal attachment to the host cell and virulence [30,32].

Other conserved cell-surface proteins that have been assayed as vaccine candidates include PiuA, PiaA, PcpA, PhtD and RrgB (recombinant protein from Pilus 1) [23].

#### 1.2.1. Preclinical Studies

Most of these proteins were assayed alone or in combination in lab animals. Many have shown potential as future candidates for independent serotype vaccines [23,25].

For example, Cecchini and coworkers reported a novel serotype-independent vaccine against *S. Pneumoniae* (PnuBioVax; PBV) composed of PspA, Ply (PlyD6), Pilus 1, and other virulence factors obtained from a modified TIG4 serotype culture [40]. PBV showed high levels of IgG antibodies against these proteins in immunized rabbits. The results suggest that PBV vaccination generates antibodies that have multiple mechanisms of action that provide effective protection against pneumococcal infection, independently of the serotype [41].

#### 1.2.2. Clinical Studies

Some of the formulations have previously been proven in children. In 2017, Odutola and collaborators described a two-part study conducted with a mixture of antigens to 10 pneumococcal serotype-specific polysaccharide conjugates (10 VT), combined with pneumolysin toxoid and pneumococcal histidine triad protein D (PHiD-CV/dPly/PhtD-30) [42]. The vaccine was well-tolerated by Gambian children. Furthermore, the authors assessed the efficacy of the formulations against pneumococcal nasopharyngeal carriage (NPC) prevalence in infants. Unfortunately, results did not show that the addition of pneumococcal proteins dPly and PhtD reduces the prevalence of nonserotype 10VT carriage beyond that provided by the vaccine without Ply and PhtD [42]. These results probably indicate that Ply and PhtD are not effective in relation to bacteria carriage, but PHiD-CV/dPly/PhtD-30 should not be discarded as an extended serotype vaccine for pneumonia cases.

Another trivalent recombinant candidate vaccine that comprised PhtD, Ply and PcpA was reported by Brooks and colleagues [43]. The vaccine formulations increased antibodies concentrations in adults, toddlers, and infants to all vaccine antigens. Although, in infants the use of adjuvant was necessary for a maximal immune response, the formulations were well-tolerated and no safety concerns were identified.

Some other formulations have been under evaluation in clinical trials as pneumococcal vaccine GSK2189242A of GlaxoSmithKline (GlaxoSmithKline, Brentford, UK). This vaccine comprised dPly and PhtD antigens and it is currently in Phase 2 with promising results [44].

## 2. Whole-Cell Pneumococcal Vaccine

### 2.1. Preclinical Assays

As an alternative to current vaccines, which have high production costs among other disadvantages, formulations with complete cells (whole-cell vaccines) express all protein antigens and could be less expensive than the purification of one or many of these proteins.

In relation to preclinical assays, in the last decade, inactivated whole cells of unencapsulated *S. pneumoniae* have been proposed to provide serotype-independent protection. These vaccine candidates have shown humoral and cellular immune responses against multiple antigens, giving very promising results in mouse infection-challenge models of colonization, pulmonary pneumonia, and sepsis [45,46,47,48].

Malley and coworkers tested killed, unencapsulated mutated Rx1 cells, applied intranasally with cholera toxin (CT) as an adjuvant in mice and rats. This formulation was shown to provide protection against nasopharyngeal colonization and invasive disease, using two different serotypes (6B, and 3) [45].

Some years later, the same preparation but with a nontoxic adjuvant was assayed with promising results in a mouse colonization model with serotypes 6B, 14, and 23F [49].

As in the clinical trials, the pneumococcal whole-cell vaccine (PWCV) used in most of the preclinical assays contained chemically killed *S. pneumoniae* RM200 cells [45]. The RM200 lineage is derived from the original progenitor, *S. pneumoniae* D39 [50]. Figure 1 shows the evolution of the cell from the original clinical isolation until now. Besides the mutations that prevent the expression of the capsule, RM200 also has another mutation: the replacement of the *lytA* gene with the Janus cassette autolysine to reduce virulence and improve the yield of cells; the alteration of the pneumolysin toxin gene that removes the protein’s cytolytic and complement-activating activity [51,52]. Glycoprotein PsrP, known as pneumococcal pili, and degradative zinc metalloprotease ZmpC were also absent, and there was a nonsense mutation in the *PspC* gene that removed six of the eight choline-binding domains (CBDs), possibly reducing the proportion of this protein attached to the cell surface [53,54].

In 2010, using RM200 cells, Lu and coworkers developed a subcutaneous injection vaccine, which was shown to provide protection in nasal colonization models with a serotype 6B strain and a fatal aspiration sepsis model with serotypes 3 and 5 in mice [47].

Similarly, HogenEsch and coworkers determined the optimal formulation of an aluminum-adjuvantized whole-cell pneumococcal vaccine. The authors reported that the vaccine containing aluminum phosphate (AP) adjuvant produced a higher antibody response and induced a critical IL-17 response in the prevention of nasopharyngeal colonization by *S. pneumoniae* [55].

Recently, Campos and coworkers showed that PWCV from RM200 cells was capable of inducing non-capsular antibodies to protect mice against invasive pneumococcal disease and reduce nasopharyngeal (NP) carriage via IL-17A production. The authors demonstrated that PWCV antibodies were able to bind to different encapsulated strains, activate the complement, and induce phagocytosis of pneumococcus in vitro [56].

Currently, investigations are focused on gaining a better understanding of the immunological mechanism involved in PWCV protection and providing improvements in the vaccine formulation through bacterial inactivation by gamma irradiation [48,57,58,59,60].

### 2.2. Clinical Assays

Pneumococcal whole-cell vaccine clinical trials registered in the US National Library of Medicine are supported by PATH (Program for Appropriate Technology in Health) and are in phase 1/2. Phase 1 was carried out in 2012 with the main goal being to determine if a *Streptococcus pneumoniae* PWCV (SPWCV) given with alum is safe and well-tolerated by healthy adults (NCT01537185). There were 42 participants and only one case of a serious adverse effect was registered that was unrelated to the vaccination (ruptured ectopic pregnancy with hemorrhage). Incidences of mild reactions were similar between the vaccinated and control groups [61]. Vaccine formulations contained chemically killed *S. pneumoniae* RM200 cells [45].

At the same time, immunoglobulin G (IgG) responses to SPWCV were characterized by pan proteome microarray using sera samples from 35 adult placebo-controlled phase 1 trial participants [62]. The results showed that SPWCV induced increases in the IgG binding of seventy-two functionally different proteins. The vaccine induced an IgG response in naturally immunogenic proteins expressed by the RM200. In addition, the vaccine induced specific responses to PclA, PspC, and ZmpB protein variants, whereas induced antibodies against PspA and ZmpA were recognized by a widespread set of alleles [54].

Recently, another clinical trial (phase 1/2), “A Dose-Finding Study to Assess the Safety, Tolerability, and Immunogenicity of Inactivated *Streptococcus pneumoniae* Whole-Cell Vaccine Formulated with Alum in Healthy Kenyan Young Adults and PCV-Primed Toddlers”, explored these issues further (NCT02097472). The purpose of this study was to assess the safety and tolerability of a *S. pneumoniae* whole-cell vaccine adsorbed to Alum (PATH-wSP) administered intramuscularly to 304 healthy Kenyan adults and toddlers who had been primed with a pneumococcal conjugate vaccine (PCV). Additionally, the study explored the immunoglobulin response when PATH-wSP was administered [63].

Preliminary results showed that the vaccine was well-tolerated. Subjects returned to the clinic one week after each vaccination, and at four weeks after the final vaccination, for assessment of solicited reactogenicity, targeted physical examination, vital signs, laboratory assessment, immunology testing, and evaluation of unsolicited adverse events and concomitant medication use. Serious adverse events occurred in less than 5% of adults and had no relation to the vaccination, while, adverse events occurred in around 4–12% of children, depending on the observed cohort, with most cases being unrelated to the vaccination.

Total adverse events were found in almost all participants, including control cohorts, and most cases were excluded from association with the vaccination [63].

Besides this, IgG measures were taken for the selected pneumococcal proteins in all cohorts with variable results, depending on the protein [63].

Clinical trials of effectivity of this vaccine are expected in the next few years.

## 3. Conclusions

The development of an effective pneumococcal vaccine has been a public-health topic for more than a century. The first pneumococcal vaccine was a whole-cell vaccine and, paradoxically, the next generation of vaccines could also be of the whole-cell type.

Nevertheless, there are several challenges before a protein or a whole-cell vaccines’ licensure can be addressed [28]. Correlates of protection are needed to support the licensure of vaccines, for that, a network of investigators from different research centers around the world have to deliver high-quality standardized functional immune assays, studies protocols comparing the relationship between antibody concentrations and disease risk in a non-vaccinated, naturally exposed population [64,65]. All the data must establish favorable safety and follow international quality standards according to Good Clinical Practices and the World Health Organization (WHO) recommendations. The new whole-cell vaccine is based on an unencapsulated serotype, and it is expected that it will allow the generation of cellular immunological responses and many types of protective antibodies. The combination of these advantages with the vaccine’s affordable cost makes it suitable for developing countries. Clinical trials showed very promising results.

On the other hand, given the crude nature (whole-cell), PWCV vaccine has more chances to induce a higher number of adverse reactions than purified antigens. Additionally, for vaccine manufacturers, the standardization and lot consistency could be more difficult to achieve with whole-cell vaccines than with purified antigens.

## Figures and Tables

**Figure 1 vaccines-07-00151-f001:**
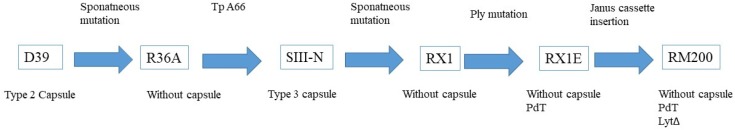
Development/genealogy of the RM200 serotype [46,47,48,50,54].

**Table 1 vaccines-07-00151-t001:** Main proteins that contribute to the virulence of pneumococcus and their functions.

Protein	Localization	Function	References
Ply	Exotoxin	Cytotoxic, TLR4 agonist, complement activator	[24,25]
PspA	Surface Protein	Inhibits the activation of the complement, binds to lactoferrin, helping adherence and protecting bacteria from lactoferrin toxic effects.	[24,25]
PspC	Surface Protein	Adhesin, inhibits activation of complement binding factor H	[24,25]
PcpA	Surface Protein	Adhesin, Choline binding protein	[26]
PhtD	Surface Lipoprotein	Adhesin, Zn binding protein	[27,28]
Autolysin	Cytoplasm/Cell Wall	Autolytic response induced during the stationary growth phase	[29]
PiuA	Surface Lipoprotein	Iron uptake ABC transporter	[25]
PiaA	Surface Lipoprotein	Iron uptake ABC transporter	[25]
PsaA	Surface Lipoprotein	Mn + 2 uptake ABC transporter adhesin	[30]
Pilus	Surface Protein	Epithelial cell adhesion	[23]

Ply (pneumolysin); PspA (pneumococcal surface protein A); PspC (pneumococcal surface protein C); PcpA (pneumococcal choline-binding protein A); PhtD (polyhistidine triad protein D); PiuA (pneumococcal iron uptake A); PiaA (pneumococcal iron acquisition A); PsaA (Pneumococcal surface antigen A); Pilus (pilus proteins).

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
