# Peer review of "Next-Generation Whole-Cell Pneumococcal Vaccine"

_vaccines, 2019, doi:10.3390/vaccines7040151_

Round 1
Reviewer 1 Report
Morais, Texeira, Suarez
Figure Caption 1. Proper references should be used that refer to reference numbers in the list of references at the end of the paper. Name and year will not suffice, I am sure this was just accidentally overlook in final drafted. .
Introduction line 4. The proper spelling of the word is “encapsulated”.
Introduction line 5. Write as, “resides”
Introduction line 5. Replace “Due to” with “Based on”. They may seem similar but yours is not the way to say it in English.
Introduction, paragraph, 3, first sentence. Your point is that immunization with a group of non-capsular immunogens can result in protection. If this is your point, don’t start by saying that CPS in the main virulence factor, this is what the capsular people have done for years and it has resulting in thinking by most people that capsule is THE virulence factor. Why don’t you start this sentence instead, by saying, “One of the most important virulence factors of . . .”?
Introduction, paragraph 4. Please note, there is a classic paper by Robert Austrian and Jerome Gold in the 1964 where they showed that penicillin can have a big effect on survival of human patients with pneumonia (they compared their pre-penicillin and post penicillin results), but especially in the elderly and very young children, the disease can progress so fast in some patients, that by the time the patients is seen and given antibiotic, it is just too late. I think this represented about 15% of the patients. This is still true, which is why even in developed countries, like yours, infections with pneumococci still cause a significant level of mortality even in treated patients. THUS, there are two reasons for a good vaccine. One is that antibiotics cannot protect everyone from even the susceptible pneumococci, and because a significant fraction of pneumococci are not antibiotic resistant. I am sending the PDF of this paper, because it may be difficult to get.
Section 1.1 paragraph 4, last sentence. This sentence would be more readily interpretable if the word “unconjugated” is inserted between “only” and “polysaccharide”.
Section 1.2, first sentence. Outermost is a poor description because actual data on this is only now becoming possible. What is known, is that CPS is the most surface accessible molecule yet characterized on the pneumococcus. Moreover, accessibility is what is probably what is important to the immune system. Fewer people will argue with you, if you say most surface accessible.
Section 1.2, line 3. Most capsular types are covalently bonded to the cell wall but a few are not. These include type 3, which is among the capsular types that are highly virulent in man. Thus, you can not ignore this. I suggest putting a period after “oligosaccharides”. Then start a new sentence and say, “Capsules of most capsular types are covalently bonded. . . “
Section 1.2, line 3. Many of the protein molecules you talk about later have functions that interfere with various host protection mechanisms. Your sentence, however, makes it sound like if pneumococci have a capsule they are totally protected from the host. If that were true, we would not observe that patients with pneumonia or bacteria generally do not die. It would be better to say that “CPS helps protect pneumococci from several of the host defense mechanisms.” If you take this (more correct) approach, then it is much easier to convince the reader that the pneumococcal proteins that you list below may have a role in pneumococcal disease.
Section 1.2, paragraph 2, line 3. Replace “progressively” with “extensively”.
Table 1is good, and lists enough proteins to make your point. If you were tempted to add another protein, I would suggest PcpA, since if is important for lung infection and sepsis.
Page 2, next to the last paragraph, last line. The word assayed not the best word you could use here. You might end this sentence with “. . . to be used in experimental vaccines”.
Page 2, last paragraph, next to the last sentence. PspA does have a variable N-terminal region, but it is also highly cross-protective and his three main families which cover virtually all pneumococci. Probably, its high exposure is why is it variable, but its exposure also makes if one of the proteins that shows at least some protection against most mouse virulent capsular types. So I make a self-serving suggestion, which you are free to ignore: say “. . . has a variable, but cross-protective, N-terminal . . .”
I have a feeling that the introduction may have been written last, because the descriptions of the vaccines in humans have clearly been worked on extensively. The text flows well and I found no places where changes in sentences could improve the paper.
It is a classic paper, but the father of the field of modern pneumococcal vaccines. It has information they may want to cite.

Author Response
Response to Reviewer 1 comments
We thank the reviewer for his valuable comments. We have added all the suggestions and added them in blue color in the revised manuscript and we have also followed point by point his advice to improve the quality of the manuscript.
Figure Caption 1. Proper references should be used that refer to reference numbers in the list of references at the end of the paper. Name and year will not suffice, I am sure this was just accidentally overlook in final drafted
1- Response: It have been done.
Introduction line 4. The proper spelling of the word is “encapsulated”
2- Response: It have been done
3- Introduction line 5. Write as, “resides”
3-Response: It have been done
4 - Introduction line 5. Replace “Due to” with “Based on”. They may seem similar but yours is not the way to say it in English
4-Response: It have been done
5- Introduction, paragraph, 3, first sentence. Your point is that immunization with a group of non-capsular immunogens can result in protection. If this is your point, don’t start by saying that CPS in the main virulence factor, this is what the capsular people have done for years and it has resulting in thinking by most people that capsule is THE virulence factor. Why don’t you start this sentence instead, by saying, “One of the most important virulence factors of
5- Response: It have been done
6- Introduction, paragraph 4. Please note, there is a classic paper by Robert Austrian and Jerome Gold in the 1964 where they showed that penicillin can have a big effect on survival of human patients with pneumonia (they compared their pre-penicillin and post penicillin results), but especially in the elderly and very young children, the disease can progress so fast in some patients, that by the time the patients is seen and given antibiotic, it is just too late. I think this represented about 15% of the patients. This is still true, which is why even in developed countries, like yours, infections with pneumococci still cause a significant level of mortality even in treated patients. THUS, there are two reasons for a good vaccine. One is that antibiotics cannot protect everyone from even the susceptible pneumococci, and because a significant fraction of pneumococci are not antibiotic resistant. I am sending the PDF of this paper, because it may be difficult to get.
6- Response
We would like to thank Rev 1 again for sending us such a precious article. Please find the changes made in blue color in the manuscript.
7 Section 1.1 paragraph 4, last sentence. This sentence would be more readily interpretable if the word “unconjugated” is inserted between “only” and “polysaccharide”.
7- Response: It have been done
8- Section 1.2, first sentence. Outermost is a poor description because actual data on this is only now becoming possible. What is known, is that CPS is the most surface accessible molecule yet characterized on the pneumococcus. Moreover, accessibility is what is probably what is important to the immune system. Fewer people will argue with you, if you say most surface accesible
8- Response: Please find all the text changes in the revised manuscript in blue color
9- Section 1.2, line 3. Most capsular types are covalently bonded to the cell wall but a few are not. These include type 3, which is among the capsular types that are highly virulent in man. Thus, you can not ignore this. I suggest putting a period after “oligosaccharides”. Then start a new sentence and say, “Capsules of most capsular types are covalently bonded. . . “
9- Response: It have been done
10- Section 1.2, line 3. Many of the protein molecules you talk about later have functions that interfere with various host protection mechanisms. Your sentence, however, makes it sound like if pneumococci have a capsule they are totally protected from the host. If that were true, we would not observe that patients with pneumonia or bacteria generally do not die. It would be better to say that “CPS helps protect pneumococci from several of the host defense mechanisms.” If you take this (more correct) approach, then it is much easier to convince the reader that the pneumococcal proteins that you list below may have a role in pneumococcal disease.
10- Response: It have been done
11- Section 1.2, paragraph 2, line 3. Replace “progressively” with “extensively”.
11- Response: It have been done
12- Table 1is good, and lists enough proteins to make your point. If you were tempted to add another protein, I would suggest PcpA, since if is important for lung infection and sepsis.
12- Response: It have been done
13- Page 2, next to the last paragraph, last line. The word assayed not the best word you could use here. You might end this sentence with “. . . to be used in experimental vaccines”.
13- Response: It have been done
14- Page 2, last paragraph, next to the last sentence. PspA does have a variable N-terminal region, but it is also highly cross-protective and his three main families which cover virtually all pneumococci. Probably, its high exposure is why is it variable, but its exposure also makes if one of the proteins that shows at least some protection against most mouse virulent capsular types. So I make a self-serving suggestion, which you are free to ignore: say “. . . has a variable, but cross-protective, N-terminal . . .”
14- Response: It has been changed. Please see the revised manuscript blue text.
15- I have a feeling that the introduction may have been written last, because the descriptions of the vaccines in humans have clearly been worked on extensively. The text flows well and I found no places where changes in sentences could improve the paper.
It is a classic paper, but the father of the field of modern pneumococcal vaccines. It has information they may want to cite.
15-Response: Thank the review for these valuables corrections.
Reviewer 2 Report
This review manuscript makes the argument that pneumococcal whole cell vaccines (WCVs) may represent a superior alternative to pneumococcal conjugate vaccines (PCVs) for the prevention of pneumococcal diseases. The paper is important because enthusiasm for non-PCV pneumococcal vaccines seems to be limited among vaccine manufacturers and funding agencies. The authors make some important points but there are also some gaps that in my opinion would considerably improve the manuscript.
Specific comments
Although this manuscript is focused on WCVs, it is also important to acknowledge both the success and limitations of PCVs i.e. a brief description of the efficacy and effectiveness of PCVs, the issue of serotype replacement and that despite the complexity in the manufacturing processes the cost of PCVs has come down to around $3/dose for low income countries due to the pneumococcal vaccine advance market commitment. Under section 1.2, Pneumococcal Antigens, it would be helpful to separate out preclinical from clinical studies. The authors should add reference to the Sanofi work on their trivalent vaccine that comprised PhtD, Ply and PcpA given that this candidate advanced into clinical evaluation in multiple studies (Brooks WA et al. (2015) Safety and immunogenicity of a trivalent recombinant PcpA, PhtD, and PlyD1 pneumococcal protein vaccine in adults, toddlers, and infants: A phase I randomized controlled study. 33(36):4610-7). There should also be reference to the more recent data on the GSK candidate (Ply plus PhtD) where data has been presented in abstract form (WSPID 2017 and ISPPD 2018) regarding their otitis media study in Navajo infants. The authors should discuss some of the challenges with development of protein and whole cell vaccines. This includes clinical proof-of-concept, licensure studies and endpoints given the now broad implementation of PCVs and the need to establish correlates of protection. Moreover, there are additional challenges for WCVs from a manufacturing perspective (characterization for potency and manufacturing consistency given the crude nature of WCVs) and a reactogenicity standpoint (e.g. with whole cell pertussis vaccine). I suggest the addition to Table 1 of two key pneumococcal proteins that have made it to clinical evaluation, namely PhtD and PcpA.
Author Response
We thank the reviewer for his valuable comments, we have followed all the suggestions in order to improve the manuscript. We have added them in blue color in the revised manuscript, trying to fill the gaps that the Reviewer found were missing.
1-Although this manuscript is focused on WCVs, it is also important to acknowledge both the success and limitations of PCVs i.e. a brief description of the efficacy and effectiveness of PCVs, the issue of serotype replacement and that despite the complexity in the manufacturing processes the cost of PCVs has come down to around $3/dose for low income countries due to the pneumococcal vaccine advance market commitment
Response: Please find the following answer at the end of the Introduction in the revised manuscript.
“The introduction conjugate pneumococcal vaccines (PCVs) have been widely adopted to be used in children in both high and low-income countries. Recently, Chen and collaborators reported a global modelling analysis of the effect and cost-effectiveness of pneumococcal conjugate vaccination [17]. The authors concluded that these PCVs show established cost-effective protection against pneumococcal diseases in various countries in terms of lives saved and disability averted.
Although vaccination with PCVs can prevent infections due to multiple serotypes of pneumococcus, the World Health Organization (WHO) reported that the universal use of PCVs have provided an opportunity for non-vaccine serotypes circulation and serotype replacement to cause an increased proportion of pneumococcal disease in both children and adults [18,19]
However, despite the persistent circulation of vaccine serotypes and serotype replacement and the complexity in the manufacturing processes, the cost of PCVs has decreased in low-income countries due to the pneumococcal vaccine advance market commitment [17]”.
Under section 1.2, Pneumococcal Antigens, it would be helpful to separate out preclinical from clinical studies. The authors should add reference to the Sanofi work on their trivalent vaccine that comprised PhtD, Ply and PcpA given that this candidate advanced into clinical evaluation in multiple studies (Brooks WA et al. (2015) Safety and immunogenicity of a trivalent recombinant PcpA, PhtD, and PlyD1 pneumococcal protein vaccine in adults, toddlers, and infants: A phase I randomized controlled study. 33(36):4610-7). There should also be reference to the more recent data on the GSK candidate (Ply plus PhtD) where data has been presented in abstract form (WSPID 2017 and ISPPD 2018) regarding their otitis media study in Navajo infants. The authors should discuss some of the challenges with development of protein and whole cell vaccines. This includes clinical proof-of-concept, licensure studies and endpoints given the now broad implementation of PCVs and the need to establish correlates of protection. Moreover, there are additional challenges for WCVs from a manufacturing perspective (characterization for potency and manufacturing consistency given the crude nature of WCVs) and a reactogenicity standpoint (e.g. with whole cell pertussis vaccine). I suggest the addition to Table 1 of two key pneumococcal proteins that have made it to clinical evaluation, namely PhtD and PcpA.
Reponse: Please find the following answer at the end of the Section 1.1.2 Clinical Studies in the revised manuscript.
Another trivalent recombinant candidate vaccine that comprised PhtD, Ply and PcpA has reported by Brooks and colleagues[40]. The vaccine formulations increased antibodies concentrations in adults, toddlers, and infants to all vaccine antigens. Although, in infants the use of adjuvant was necessary for a maximal immune response, the formulations were well tolerated and no safety concerns were identified.
Some other formulations have been under evaluation in clinical trials as pneumococcal vaccine GSK2189242A of GlaxoSmithKline. This vaccine comprised dPly and PhtDantigens and it is currently in Phase 2 with promising results [41]
Response: The proteins have been added to Table 1.
Round 2
Reviewer 2 Report
I think the manuscript is suitably improved and appropriate for publication with minor English language improvements.